# What Can Glioma Patients Teach Us about Language (Re)Organization in the Bilingual Brain: Evidence from fMRI and MEG

**DOI:** 10.3390/cancers13112593

**Published:** 2021-05-25

**Authors:** Ileana Quiñones, Lucia Amoruso, Iñigo Cristobal Pomposo Gastelu, Santiago Gil-Robles, Manuel Carreiras

**Affiliations:** 1Neurobiology of Language Group, Basque Center on Cognition, Brain and Language (BCBL), 20009 Donostia-San Sebastián, Spain; l.amoruso@bcbl.eu (L.A.); m.carreiras@bcbl.eu (M.C.); 2IKERBASQUE, Basque Foundation for Science, 48009 Bilbao, Spain; 3Department of Neurosurgery, Hospital Cruces, 48903 Bilbao, Spain; INIGOCRISTOBAL.POMPOSOGASTELU@osakidetza.eus; 4BioCruces Research Institute, 48015 Bilbao, Spain; santigilrob@yahoo.es; 5Department of Neurosurgery, Hospital Quironsalud, 28223 Madrid, Spain; 6Department of Basque Language and Communication, University of the Basque Country, UPV/EHU, 48940 Bilbao, Spain

**Keywords:** glioma patients, functional mapping, fMRI, MEG, bilingualism

## Abstract

**Simple Summary:**

Low-grade glioma (LGG) patients constitute an ideal in vivo pathological model to investigate cerebral neuroplasticity associated with major architectural disruption to the language network. Bilingual LGG patients offer a unique opportunity to study the neural capacity to negotiate L1 and L2 processing before and after the resection of critical language hubs. By combining the spatial resolution of fMRI with the temporal resolution and oscillatory information provided by MEG, we mapped the language network and its functional (re)organization in five Spanish–Basque bilingual patients. Both techniques provide converging evidence that different reshaping patterns occur for L1 and L2 after tumor resection. These changes affect not only language-specific nodes, but also areas associated with executive control mechanisms, underscoring the need for multilingual intraoperative approaches. Understanding neural (re)organization in the bilingual brain is crucial for preserving language function by means of personalized surgical interventions and rehabilitation strategies based on the patient’s linguistic profile.

**Abstract:**

Recent evidence suggests that the presence of brain tumors (e.g., low-grade gliomas) triggers language reorganization. Neuroplasticity mechanisms called into play can transfer linguistic functions from damaged to healthy areas unaffected by the tumor. This phenomenon has been reported in monolingual patients, but much less is known about the neuroplasticity of language in the bilingual brain. A central question is whether processing a first or second language involves the same or different cortical territories and whether damage results in diverse recovery patterns depending on the language involved. This question becomes critical for preserving language areas in bilingual brain-tumor patients to prevent involuntary pathological symptoms following resection. While most studies have focused on intraoperative mapping, here, we go further, reporting clinical cases for five bilingual patients tested before and after tumor resection, using a novel multimethod approach merging neuroimaging information from fMRI and MEG to map the longitudinal reshaping of the language system. Here, we present four main findings. First, all patients preserved linguistic function in both languages after surgery, suggesting that the surgical intervention with intraoperative language mapping was successful in preserving cortical and subcortical structures necessary for brain plasticity at the functional level. Second, we found reorganization of the language network after tumor resection in both languages, mainly reflected by a shift of activity to right hemisphere nodes and the recruitment of ipsilesional left nodes. Third, we found that this reorganization varied according to the language involved, indicating that L1 and L2 follow different reshaping patterns after surgery. Fourth, oscillatory longitudinal effects were correlated with BOLD laterality changes in superior parietal and middle frontal areas. These findings may reflect that neuroplasticity impacts on the compensatory involvement of executive control regions, supporting the allocation of cognitive resources as a consequence of increased attentional demands. Furthermore, these results hint at the complementary role of this neuroimaging approach in language mapping, with fMRI offering excellent spatial localization and MEG providing optimal spectrotemporal resolution.

## 1. Introduction

A remarkable feature of the central nervous system is its inherent capacity to dynamically reorganize its structure and function depending on the environment [1,2,3,4,5,6]. This capacity for neuroplasticity can give rise to changes in both healthful (e.g., acquisition of a new language; [7,8]) and pathological (e.g., presence of brain lesions; [4,5]) contexts. Our understanding of how cognitive processing is influenced by neuroplastic mechanisms induced by a focal and sudden disruption of a specific cognitive network is predominantly based on research with stroke and epilepsy cases [9,10,11,12,13]. These pathological models have revealed various plasticity patterns, including functional reorganization in perilesional regions, ipsilesional recruitment of long-distant areas, and engagement of contralesional homologs. These reorganization mechanisms ensure a flexible redistribution across the system that can support cognitive recovery [14,15]. However, with brain tumor patients, the picture becomes more complex. Low-grade glioma (LGG) is a progressive and invasive disease that affects cognitive faculties. The slow growth of this type of lesion allows the brain to reorganize its structure and functions, delaying the onset of cognitive symptoms [4,5,16,17]. Thus, a brain tumor impinging on areas contributing to a specific cognitive function, and its surgical resection, does not necessarily imply cognitive impairment related to that region’s functionality [18,19].

The current study is aimed at investigating neuroplasticity in a pathological situation where the brain is forced to find adaptive solutions to a major change in the architecture of the language connectome: the surgical resection of a language hub. To cope with the presence of a tumor, neuroplasticity mechanisms transfer linguistic functions from damaged to healthy areas unaffected by the tumor. This phenomenon has been reported in monolingual patients, but is understudied in populations that speak more than one language. As the world becomes increasingly bilingual, understanding whether an individual’s languages engage similar or different neural networks—and whether damage to such language hubs would result in diverse recovery patterns—requires further attention.

LGG patients constitute an ideal in vivo pathological model to investigate cerebral neuroplasticity associated with major architectural disruption to the language network. Bilingual LGG patients offer a unique opportunity to study the neural capacity to negotiate first and second language information after resection of critical language hubs. Understanding neural reorganization in bilingual brain tumor patients is critical for preserving these areas and thus preventing involuntary pathological symptoms following resection. Indeed, from a clinical standpoint, if a patient speaks multiple languages, functional mapping of each language should be performed [20,21].

Neurologists and neurosurgeons typically use non-invasive neuroimaging techniques to diagnose tumors, plan interventions, and design neuro-rehabilitation strategies for patients. The standard clinical neuroimaging protocol includes structural (<1 mm^3^ resolution) and task-related functional (~2 mm^3^ resolution) Magnetic Resonance Imaging (fMRI), which provides precise information on the different types of soft tissues that compose the structural and functional human brain in vivo. However, even though fMRI studies have played a pivotal role in advancing our understanding of the functional and structural architecture of the language network and its reorganization in the lesioned brain, the temporal resolution of fMRI is limited (i.e., one volume every ~2 s). This aspect becomes critical when considering linguistic processing, which occurs on the subsecond timescale. Ideally, neuroplasticity should also be examined with high-temporal resolution techniques that are capable of capturing oscillatory fluctuations associated with language function in real time [22]. Neurophysiological techniques such as electroencephalography (EEG) and magnetoencephalography (MEG) meet these requirements, as they can measure neuronal activity with millisecond resolution, offering new insights into brain plasticity. Furthermore, these techniques allow researchers to capture the spectrotemporal fingerprints of distinct language operations that would remain undetected using traditional fMRI approaches. Indeed, it has been suggested that spectro-temporal information, namely how neural oscillations evolve over time at different frequency bands, reflects network dynamics supporting cognition [23].

In the current study, we combined the high spatial resolution of fMRI with the high temporal resolution and oscillatory information provided by MEG to map the language network and its functional (re)organization in the bilingual brain, both prior to and following tumor-removal surgery. This multimethod approach is certainly a major methodological challenge for clinical neuroscience, which provides invaluable information about neuroplasticity, considering the whole brain as a set of dynamic networks [24]. Specifically, we tested five Spanish–Basque bilingual patients with low-grade gliomas before and four months after surgery for tumor resection. This pathological model allows evaluating their neural capacity to negotiate L1 and L2 linguistic information after the resection of critical language hubs. This longitudinal approach (post- vs. pre-surgery) granted us an exceptional opportunity to estimate functional neuroplasticity at the individual level, considering the patient’s cognitive status both prior to and following tumor removal surgery. We were able to investigate the impact of slow-growing tumors on the architecture and functionality of the language network, considering each patient as a unique and informative case. The longitudinal design also reduced confounds due to interindividual variability by making each patient his/her own control. It has been shown that longitudinal studies often exhibit less variability and better statistical power than cross-sectional ones [25]. Furthermore, in pathological populations, identifying individual phenotypes is critical for designing successful intervention and rehabilitation strategies customized for each patient.

Overall, based on the previous literature, we hypothesized the existence of similar plasticity mechanisms as those observed in stroke patients [9,10,11,12,13], namely the recruitment of ipsilesional healthy areas and contralateral homologs. Specifically, we expected that changes induced by these mechanisms would be evident in a longitudinal fashion [26]. To the best of our knowledge, no study has previously tracked plastic changes in the bilingual brain through fMRI and MEG language mapping using a longitudinal approach. Nevertheless, based on studies investigating bilingual language control, we predicted a compensatory upregulation of domain-general areas involved in inhibition and attentional control mechanisms [27]. As the superior parietal gyrus, the cingulate cortex, and the middle/superior frontal areas are at the heart of the inhibition and attentional control network, we specifically expected a language effect on some of these regions [27,28,29], which might have a counterpart on cross-language alpha-power changes [30,31,32].

## 2. Materials and Methods

### 2.1. Participants

Five Spanish–Basque bilingual patients with LGGs took part in this study. They all had normal hearing and normal or corrected to normal vision. Individual patients’ demographics, lesions, and clinical characteristics are summarized in Table 1 (see Figure 1 for 3D reconstructions of the lesions). Patients were recruited at the Hospital Universitario Cruces Bilbao (Spain), where they received their diagnosis and underwent awake brain surgery for tumor resection (MD. Ph.D. Santiago Gil-Robles—Head of the Neurosurgery Department of the Hospital Universitario Quirónsalud Madrid, Spain—, and M.D. Iñigo Pomposo Gastelu—Head of the Neurosurgery Department of the Hospital Universitario Cruces Bilbao, Spain—were the neurosurgeons in charge). They all had been referred to the neurosurgery service as asymptomatic patients. The initial neurological examinations at the hospital revealed no motor, somatosensory, or linguistic deficits. Patients were recorded in two sessions: the first session occurred one week before surgery and the second one, approximately four months after surgery. In each session, behavioral, MRI, fMRI, and MEG data were collected.

### 2.2. Cognitive Assessment

Participants were assessed using a set of standardized neuropsychological and behavioral tests. These tests included measures of general cognitive status [33], verbal and non-verbal intelligence [34], and language production [35]. These measures provided us with a detailed cognitive characterization of each individual. Tests per cognitive domain are listed in Table 2.

### 2.3. Bilingual Picture-Naming Test Used to Characterize the Language Network

The functional organization of the language network was tested using MULTIMAP, a multilingual picture-naming task for mapping eloquent areas during awake surgeries [36]. This event-related task includes classical object-naming (nouns) and action-naming (verbs); participants view a picture and name the object or associated action depicted, respectively (see Figure 2). This type of task has been extensively used to investigate the brain mechanisms underlying language production as well as to assess language-function integrity in pathological populations [26,37]. In order to avoid BOLD changes associated with task switching and differences in the attentional burden across lexical categories (i.e., object naming and verb generation), object-naming and action-naming were tested in different blocks.

MULTIMAP consists of a database of standardized color-pictures representing both objects and actions. These images have been tested for name agreement with speakers of various languages, including Spanish and Basque, and controlled for relevant linguistic features in cross-language combinations. For the purpose of the current study, we used a subset of 88 drawings, including 44 objects and 44 actions. Target words were matched on frequency, number of orthographic neighbors, and length (i.e., 5–8 characters). In addition, the stimuli were controlled for visual complexity, familiarity, and name agreement (i.e., higher than 80%). Values for imageability and concreteness were high for both nouns (mean imageability = 6.20, SD = 0.37; mean concreteness = 5.88, SD = 0.47), and verbs (mean imageability = 5.25, SD = 0.57; mean concreteness = 4.73, SD = 0.65).

The stimuli were visually presented in the center of the screen for one second, followed by an inter-stimulus interval. While the mean ISI for the fMRI version of the task was 5.46 s (varying between two and eight seconds), the ISI for the MEG varied randomly between 2 and 3 seconds. Above each object, we added the text *“Esto es…”* or *“Hori da…”*—“This is…” in Spanish and Basque, respectively—to force participants to produce a short sentence that agreed in number and gender with the target noun. In the case of the action pictures, we included a pronomial phrase to be used as the subject of the sentence, that is, either *“Él…” / “Ella…”* or *“Hark…”*—“He…” or “She…” in Spanish and Basque, respectively. This introductory text was used as a cue for the production of a sentence that started with the given subject and had a finite verb form in the 3^rd^ person singular. Participants’ responses were recorded in order to estimate accuracy and reaction time per trial. We used MATLAB version 2012b and Cogent Toolbox [38] to present the images (the stimuli, the Matlab script, and its compiled version are available for use at [39].

### 2.4. MRI Data Acquisition

All participants underwent an MRI session in a 3T Siemens Magnetom Prisma Fit scanner (Siemens AG, Erlangen, Germany). High-resolution T1- and T2-weighted images were acquired with a 3D ultrafast gradient echo (MPRAGE) pulse sequence using a 64-channel head coil with the following acquisition parameters for T1: 176 contiguous sagittal slices; voxel resolution 1 × 1 × 1 mm^3^; Repetition Time (RT) = 2530 ms, Echo Time (ET) = 2.36 ms; Image columns = 256; Image rows = 256; flip angle (Flip) = 7° and for T2: 176 contiguous sagittal slices; voxel resolution 1 × 1 × 1 mm^3^; RT = 3390 ms, ET = 389 ms; Image columns = 204; Image rows = 256; Flip = 120°. For each patient, the origin of the T1/T2 weighted images (pre and post) was set to the anterior commissure. After that, the four structural images were co-registered. A total of 368 echo-planar functional images were recorded using the following parameters: number of slices = 72; voxel size = 2 mm^3^ isotropic; ET = 29 ms; repetition time RT = 1.8 s; Field of View (FoV): 192 mm; matrix = 864 × 864; Flip = 73 degrees; acceleration factor = 1; Echo spacing = 10.42 ms. In order to guarantee steady-state tissue magnetization, the first six volumes of each functional run were discarded.

### 2.5. GLM-Based Functional MRI Data Analysis

Functional event-related data were analyzed using SPM12 and related toolboxes [40]. Raw functional scans were slice-time corrected taking the middle slice as the reference, spatially realigned, unwarped, co-registered with the anatomical T1, and normalized to MNI space using the unified normalization segmentation procedure. Global effects were then removed using a global signal regression analysis [41], after which the data were smoothed (8 mm^3^ isotropic Gaussian kernel) and high-pass filtered (128 s cut-off period).

Single case statistical parametric maps were estimated following a robust regression with weighted least-squares [42] including a regressor for each condition and the six motion-correction parameters as regressors of non-interest. The regressors of interest comprised the onset of each sentence trial and the different experimental manipulations—i.e., Spanish objects, Spanish verbs, Basque objects, Basque verbs—before and after surgery. For the purpose of the current experiment, we focused our statistical analyses on individual longitudinal variations across languages. For this reason, we did not distinguish between objects and verbs. The effects of Language (Spanish and Basque) and Time (post- and pre-surgery) were estimated following a single-case approach. Individual effects were tested using a threshold of *p* < 0.001 uncorrected with a voxel extent higher than 50 voxels; then, the statistical table was analyzed and the *p* and *k* values were corrected such that only those peaks or clusters with a *p*-value corrected for multiple comparisons using false discovery rate (FDR) [43] were considered to be significant (*p* < 0.05). All local maxima are reported in MNI coordinates [44].

### 2.6. Laterality Index Estimation Based on fMRI Data

We tested whether speech production across languages and time points produced similar or different patterns of brain lateralization. Laterality indexes (LI) for the critical contrasts were estimated per participant and region following the procedure included in the Laterality Index SPM Toolbox [45]. Following a threshold-independent bootstrapping approach, LI was estimated as LI = (left − right)/(left + right), resulting in positive values for left-dominance and negative values for right-dominance [46]. This involves iterative resampling and estimation of LIs across multiple threshold levels for all possible right/left sample combinations. In order to reduce the effect of outliers, trimmed means taken from the middle 50% of the *t*-value distribution were used as the final LI scores [46]. The 54 ROIs used as spatial constraints were built in MNI space using the AAL atlas.

### 2.7. MEG Data Acquisition

MEG signals were recorded in a magnetically shielded room by means of an Elekta Neuromag system (360-channels, Helsinki, Finland). Two pairs of electrodes located in the external chanti of each eye and above and below the right eye were used to examine participants’ eye movements. Electrocardiographic (ECG) activity was monitored by positioning one electrode below the right clavicle and another electrode under the left rib bone. Continuously recorded MEG signals were obtained at a 1 kHz sampling rate and filtered online between 0.1–330 Hz. The position of the participant’s head inside the helmet was tracked during the whole recording session with five head position indicator (HPI) coils. Coil location was digitalized relative to the nasion as well as left and right pre-auricular anatomical fiducials (FastrakPolhemus, Colchester, VA, USA). Additionally, 200 points distributed over the scalp were digitalized to align the MEG sensor coordinates space to the participant’s T1 high-resolution structural MRI.

### 2.8. MEG Data Pre-Processing

Continuous MEG data were preprocessed offline by means of the spatiotemporal signal space separation (tSSS) method [47] implemented on Maxfilter 2.2 (Elekta-Neuromag, Elekta Oy, Helsinki, Finland). Briefly, tSSS removes external magnetic noise from the MEG signal, corrects for participants’ head movements, and performs bad channel interpolation. Subsequent analyses were run with the FieldTrip toolbox [48] in MatlabR2014B (The MathWorks Inc., Natick, MA, USA, 2014.). Data were downsampled to 500 Hz and segmented into trials time-locked to picture presentation, ranging from 500 ms before to 1000 ms after image onset.

A semi-automatic procedure was used to remove those trials containing muscular, jump and flat signal-related artifacts. Furthermore, a fast independent component analysis (ICA) was employed to correct for EOG and ECG artifacts [49].

### 2.9. Selection of Frequency-Band and Time-Window

Previous M/EEG evidence suggests that power changes in low-frequency bands reflect the retrieval of lexical–semantic information [26,50,51,52,53]. Thus, here, we focused on theta (4–8 Hz), alpha (8–12 Hz), and beta (13–28 Hz) oscillations. The time window for our TFR analysis was chosen based on methodological constraints imposed by the overt nature of the task. Indeed, previous studies show that, when considering speech production tasks, artifact-free neural recordings can be measured up to ~400 ms after stimulus presentation [54]. Thus, based on this evidence, we decided to focus our analysis on the 0–500 ms time window after picture onset. Of note, this time window allowed us to tackle the critical stages of speech production, namely, conceptualization and lexical selection processes [55,56].

### 2.10. MEG Sensor-Level Analysis

Time–frequency representations (TFR) were estimated from the clean MEG trials in the 1–30 Hz frequency range. TRFs were calculated using Hanning tapers and a fixed time window of 500-ms length, resulting in a 2-Hz frequency resolution. Power was separately estimated for each orthogonal direction of a gradiometer pair and further combined, for a total of 102 measurement sensors. Power changes were expressed relative to a ~500 ms baseline period before stimulus presentation. Differences in spectral power between conditions were calculated using a cluster-based permutation approach [57]. Briefly, the test uses a cluster-based correction for multiple comparisons while maintaining sensitivity. For the longitudinal contrasts (post- vs. pre-surgery in Spanish and Basque), we averaged over frequency bins of interest (4–8 Hz for the theta band, 8–12 Hz for the alpha band, and 13–28 Hz for the beta band) and 102 sensors (i.e., combined gradiometers) in two separate time windows, namely: from 100 to 350 ms and from 350 to 500 ms after picture onset. The permutation *p*-value was obtained with the Monte-Carlo method, using 1000 random permutations. The alpha threshold was a *p*-value below 5% (two-tailed).

### 2.11. MEG–fMRI Correlational Analysis

We conducted a correlational analysis to explore potential relationships between BOLD and neurophysiological MEG activity. First, we calculated the mean of theta (4–8 Hz) and alpha (8–12 Hz) power within the significant sensors (i.e., combined gradiometers) in the time windows highlighted by the cluster-based permutation test for each language (i.e., Spanish and Basque) and stage (i.e., before and after tumor resection). Second, we estimated power indexes by subtracting post- and pre-surgery mean values to capture the longitudinal effects highlighted by the clusters. In other words, these indexes reflected how theta and alpha power longitudinally changed due to plasticity induced by the surgical resection of the tumor. Finally, these indexes were correlated with the longitudinal laterality indexes derived from the fMRI analysis. We limited our correlational analyses to the 18 cortical regions defined as the language network and the executive-control network in the functional network-based parcelation proposed by [29] (i.e., out of 54 ROIs). Pearson correlations between MEG and fMRI longitudinal measures were performed. Bonferroni correction was applied to control for the probability of committing a type I error adjusting the alpha value for the number of statistical tests (i.e., 0.05/18 = 0.003).

## 3. Results

### 3.1. GLM-Based fMRI Longitudinal Effects

Overall, widespread patterns of activation appeared for Basque and Spanish when we estimated the BOLD response associated with language production before surgery. Activation spread along the ventral and dorsal pathways, including areas in both hemispheres. In general terms, the left inferior frontal gyrus, left insula, right and left middle, and superior temporal gyri, right and left cingulate cortex, right inferior parietal lobe, left supramarginal gyrus, and left supplementary motor area exhibited significantly greater activity than a fixation baseline condition (see Figure 3A for the superimposition across participants). Despite some differences across participants, longitudinal contrasts (post- vs. pre-surgery) within each language, as tested in individual GLM analyses, showed an increase in the activation level and number of activated nodes in all five patients. Ipsilesional activation of distant areas and the recruitment of contralesional homologs were observed in all patients (Figure 3).

In order to statistically test for change in brain lateralization, we estimated laterality indexes for 54 functional regions taken from the AAL atlas [59]. As can be observed in Figure 3B, lateralization indexes spanned the full range of possible values between –1 (fully right-lateralized) and 1 (fully left-lateralized). Despite the fact that patterns of brain lateralization were highly variable across participants (individual LI data per language are available at [60]), a critical finding emerged: functional reshaping across all five bilingual patients included recruitment of new regions not only in the contralateral but also in the ipsilateral hemisphere. Figure 4 shows plots of LI longitudinal changes for each of the five patients separately for Spanish and Basque. The middle black circle represents no activation (either on the left or the right hemisphere), while the outer circle represents left and the inner circle right laterality. This change in lateralization patterns across sessions (post- vs. pre-surgery) varied as a function of the language: larger effects were found for L2 (i.e., Basque) than the L1 (i.e., Spanish).

A detailed characterization of fMRI responses per Language (Spanish and Basque) and Time (pre- and post-surgery) is presented in Figure 5 for a prototypical patient. GLM parameters were estimated using a robust regression with weighted least-squares [42]. The effects of Language (Spanish and Basque) and Time (post- and pre-surgery) were estimated following a single-case approach. Only those peaks or clusters with *p*-values corrected for multiple comparisons using family-wise error rate (FWER) [61] or false discovery rate (FDR) [43] were considered to be significant (*p* < 0.05) and were represented in the figure. In the case of this patient, the left and right middle temporal gyrus was bilaterally recruited both prior to and following tumor removal surgery, whereas the left inferior frontal gyrus was recruited only after surgery. The right part of this figure displays longitudinal laterality changes (post- vs. pre-surgery). Contrasting longitudinal effects across languages, this patient exhibited a greater increase in right lateralization for Basque than Spanish.

### 3.2. Longitudinal Sensor-Level Effects

The comparison of oscillatory activity across sessions (post- vs. pre-surgery) within each language, yielded significant early (~100–350 ms) differences in the theta frequency-band (4–8 Hz) for both Spanish and Basque (both Monte Carlo *p* = 0.004, two-tailed). Overall, power decreases were observed in right frontotemporal sensors after tumor resection for Spanish, and in left frontal sensors for Basque (see Figure 6). In addition, we found late (~350–500 ms) power increases (i.e., less desynchronization) in the alpha band (8–12 Hz) in right parietal and middle-temporal sensors for Basque only (Monte Carlo *p* = 0.004, two-tailed). Of note, these longitudinal effects observed at the group level, were largely consistent at the individual patient level (see Figure 7). Finally, no significant effects were observed for either Spanish or Basque in the beta band (13–28 Hz).

### 3.3. Results of the Correlational Analysis

Both theta and alpha oscillatory longitudinal indexes and fMRI longitudinal lateralization indexes were correlated taking into account 18 cortical regions. We found two significant negative correlations, one for Spanish and the other one for Basque, using an alpha level threshold of 0.003 (Bonferroni corrected for 18 comparisons) suggesting an interesting convergence between MEG and fMRI results. First, theta longitudinal changes were negatively related with middle frontal gyrus lateralization index in Spanish (*r* = −0.98; *p* = 0.003), reflecting that the more theta power decreases in the right hemisphere after tumor resection, the less the rightward lateralization of the middle frontal gyrus (see Figure 8). Second, in the case of Basque, alpha longitudinal changes were negatively related (*r* = −0.99; *p* = 0.001) with the longitudinal lateralization pattern of the superior parietal gyrus: the more alpha power increases in the right hemisphere after tumor resection, the less the leftward lateralization of the superior parietal. These results might indicate that post-surgery changes in theta (i.e., power decreases) and alpha (i.e., power increases) are related to the disengagement and engagement, respectively, of contralateral regions. However, this hypothesis should be tested in further studies with a larger sample.

### 3.4. Longitudinal Cognitive Recovery

Participants were longitudinally assessed using standardized neuropsychological and behavioral tests. Individual longitudinal changes are shown in Figure 9. Results showed that all the patients preserved linguistic function in both languages after surgery. In the case of the Minimental Cognitive State Examination and the Kbit, some patients performed better post-surgery than pre-surgery, while others obtained similar scores.

## 4. Discussion

In the present study, we aimed at investigating longitudinal language reorganization (pre- vs. post-surgery) in bilingual brain tumor patients by means of a combined methodological approach including fMRI, MEG, and behavioral measures. By testing Spanish–Basque bilingual LGG patients, we were able to investigate whether the longitudinal reshaping of the language system differentially affects L1 and L2 processing. Overall, four main findings can be highlighted. First, all patients preserved linguistic function in both languages after surgery, suggesting that the surgical intervention, together with the intra-operative language mapping performed during tumor resection, were successful in preserving the cortical and subcortical structures necessary to allow brain plasticity at the functional level. Second, we found a reorganization of the language network after tumor resection in both languages, mainly reflected by a shift of activity to right hemisphere language nodes and by the recruitment of ipsilesional left nodes. This suggests that similar neuroplasticity mechanisms preserve language function in bilingual patients as those observed in monolingual patients [9,10,12,13,16,18]. Third, we nonetheless observed that this reorganization varied according to the language involved, with the L1 and L2 exhibiting differential plasticity after surgery, perhaps reflecting a non-complete overlap in their neural representation. Importantly, these effects were observed using both fMRI and MEG. Specifically, when considering fMRI results, a stronger shift of activity in parietal, cingulate, and superior frontal regions toward the right hemisphere was found for Basque, as compared to Spanish. In the case of MEG results, a differential alpha oscillatory response in right parietal scalp sensors was specifically observed for Basque, with the five patients exhibiting alpha power increases after surgery irrespective of tumor location. On the other hand, theta longitudinal effects were observed for both Spanish and Basque, with power decreases after tumor resection. Fourth, the alpha power effect for Basque was correlated with BOLD rightward laterality changes in superior parietal regions. In addition, power theta changes in Spanish were related to left shifts in BOLD activity in middle frontal areas. Overall, these findings may reflect that neuroplasticity impacts on the compensatory involvement of executive control regions supporting the allocation of cognitive resources as a consequence of increased attentional demands. Indeed, the set of areas highlighted by our multimethod approach involved parietal and frontal structures previously linked to the executive control network. Overall, these results underscore the complementarity of fMRI and MEG techniques in language mapping for brain tumor patients as well as their respective advantages: spatial localization and spectro-temporal resolution.

The early neuroimaging studies of bilingualism attempted to establish whether L1 and L2 were represented in common or different territories in the brain. Empirical evidence revealed conflicting results, some of them showing a high degree of overlap between L1 and L2 [62,63,64] and others showing that each language additionally activated separate regions [65,66]. A large number of neurocognitive studies have since revealed how the dynamics of the language network change after the acquisition of a second language. These empirical findings have suggested that this neuroplastic process permeates both brain structure and function [2,3,6,67]. This process has also been explored using longitudinal approaches. Specifically, training studies of typical populations have identified a causal link between intensive training in a second language and changes in cortical brain architecture [3,68,69]. These changes included different nodes within the language circuit such as the anterior, inferior, and superior temporal gyrus, the middle and inferior frontal gyrus, premotor regions, and the cingulate cortex. Another critical aspect when considering bilingualism is how individuals manage to control for potential interferences of one language over the other. When considering speech production, a recent influential view on bilingual language control is the adaptive control hypothesis [70], which posits that bilinguals recruit domain-general executive control regions to manage competition between languages depending on the context. Thus, as in other types of cognitive effortful tasks (e.g., error monitoring, conflict resolution), control regions including the superior parietal, the prefrontal cortex, the anterior cingulate cortex, the inferior frontal gyrus, and supplementary motor areas have also been reported to be involved when bilinguals control for potential language interference [28,63,70,71,72,73].

In the case of brain tumor patients, evidence mainly comes from studies using direct cortical stimulation [21,74] during awake craniotomy for tumor resection. These studies suggest that although there are areas shared by both languages, there are also cortical territories (e.g., in frontal, parietal, and posterior temporal regions) that preferentially respond to one language or another. Network-based approaches have recently been used to investigate how the dynamics of the language network are affected when a brain lesion appears [75,76,77]. For instance, based on resting-state data, a previous study has shown changes in the functional connectivity between different contralateral and ipsilateral nodes within the language network, including the inferior frontal gyrus, superior temporal gyrus, and temporoparietal junction [75].

Here, we found that both fMRI and MEG were successful in detecting different patterns of reorganization after surgery in Spanish and Basque patients. For instance, in the LI fMRI analysis, the functional shift in activity toward the right hemisphere was greater for Basque than Spanish. In the case of the MEG TFR analysis, different spectrotemporal dynamics were observed for the two languages, with right alpha modulations only present in Basque. This latter oscillatory effect may suggest differential compensation depending on language proficiency. Indeed, right temporoparietal alpha activity has been previously related to executive control [30] and, more specifically, to language inhibition in bilingual patients [31,32]. Thus, these longitudinal alpha power increases may potentially reflect the presence of differential cognitive demands when processing L2 representations. Indeed, the location of the effect suggested by the gradiometer topographies is in line with previous MEG findings [30] and current fMRI results, showing changes in the lateralization of parietal, cingulated, and superior frontal nodes toward the right hemisphere, and also with past studies indicating that the parietal cortex of bilinguals is associated with language inhibition [78]. Furthermore, in support of this interpretation, our correlational analysis showed a strong association between alpha longitudinal changes and rightward lateralization of BOLD activity in the superior parietal cortex. Thus, these differential patterns of reorganization might indicate that, in order to preserve linguistic functions, different compensatory networks are called into play depending on language proficiency, including the engagement of executive control networks previously reported in bilingual language control of healthy individuals [70]. However, this claim needs to be tested empirically with larger samples and controlling for individual linguistic profiles (e.g., L2 proficiency, L2 age of acquisition).

Nevertheless, it is true that similarities between both languages were also found. For instance, theta power changes in frontal sensors were present irrespective of the language at use between 100 and 350 ms after picture onset. Previous studies have associated theta activity with the retrieval of lexicosemantic information during language comprehension [50,79,80] and production [51]. Some other studies, however, suggest that frontal theta oscillations constitute a hallmark signature of executive control, with increased power observed during error monitoring and when the amount of top–down control is higher due to conflict/interference [81,82,83,84]. Interestingly, a few recent studies on speech production [85] also report frontal theta activity modulations underscoring its potential involvement in a domain-general control mechanism impacting on language processing. In line with this evidence, we found a strong association between the frontal theta effect in Spanish and the BOLD laterality changes in the middle frontal gyrus, showing disengagement of right contralateral regions after tumor resection.

Indeed, the fact that increased frontal theta activity was present in both languages before surgery seems to reflect that, in the presence of the tumor, a stronger top–down controlled retrieval of lexico-semantic information was required to maintain function. It is likely that, after successful tumor resection, this compensation was no longer required as reflected in the theta power decreases and the shift back to the left hemisphere. Nevertheless, in light of the observed correlation found only in Spanish, this effect may indicate that, even though a compensatory activity was necessary for both L1 and L2, this compensation might have played a less critical role in the case of the native language (i.e., downregulation of right areas after surgery). Overall, these oscillatory effects may be plausibly underpinned by the typical neuroplasticity mechanism shown in previous studies in monolingual individuals [26] involving the additional and/or stronger recruitment of frontal ipsilesional nodes to maintain network homeostasis.

Finally, we would like to point out the advantage of combining different neurofunctional imaging methods, such as fMRI and MEG, during preoperative and postoperative assessment in the quest to understand brain plasticity. Previous literature on brain plasticity induced by pathological events, such as the tumor resections explored here, has underscored three main patterns of plasticity in terms of spatial change: the recruitment of perilesional areas, the activation of ipsilesional areas distant from the lesion, and the recruitment of contralesional homologs. All of these patterns are mainly concerned with “*where*” changes occurred in the brain, an aspect best captured by techniques with high spatial resolution such as fMRI. However, when investigating the reshaping of a complex cognitive system such as the language system, we need to have access not only to the brain region(s) involved, but also the temporal and spectral dynamics of brain activity to characterize neural change at the network level. Neurophysiological techniques such as EEG and MEG offer better temporal resolution than fMRI [66] and allow us to tap into rapid language processes occurring in a subsecond time-scale. Furthermore, brain rhythms captured by M/EEG techniques are thought to reflect communication between brain areas and thus can provide information about network dynamics. Hence, these techniques are required to have a more complete portrayal of neuroplasticity changes [22]. It has been demonstrated that fMRI results are very consistent with more direct brain mapping techniques such as intracranial evoked potentials and electrocortical mapping [86]. However, unfortunately, the BOLD response in areas surrounding a tumor is very noisy and does not reflect the neuronal signal as accurately as in healthy tissue, probably due to changes in neurovascular and metabolic coupling [87,88]. In contrast to fMRI, MEG is a suitable tool for measuring neurofunctional activity in areas surrounding a lesion. Therefore, by combining fMRI and MEG in a longitudinal fashion, we can go beyond the “*where*” and also examine the physiological mechanisms underlying “*when*” neuroplastic changes occur and “*what*” oscillatory dynamics subserve these processes.

## 5. Conclusions

Our fMRI-MEG findings suggest that language reorganization takes place in the bilingual brain after tumor resection following neuroplasticity mechanisms similar to those observed in monolingual patients (i.e., recruitment of ipsilesional and contralesional nodes). Furthermore, they show that this language rearrangement occurs in both L1 and L2, underscoring the necessity of mapping all the languages that a patient speaks. Finally, they also hint at differential post-surgery reorganization of L1 and L2 affecting not only language-specific nodes, but also areas associated with executive control mechanisms. These findings suggest that the changes required to preserve cognitive functions may trigger the compensatory engagement of different networks depending on an individual’s language experience (e.g., L2 proficiency). Overall, from a clinical standpoint, these findings help delineate personalized surgical strategies that respect a patient’s linguistic profile in order to preserve language function in an integral fashion.

## Figures and Tables

**Figure 1 cancers-13-02593-f001:**
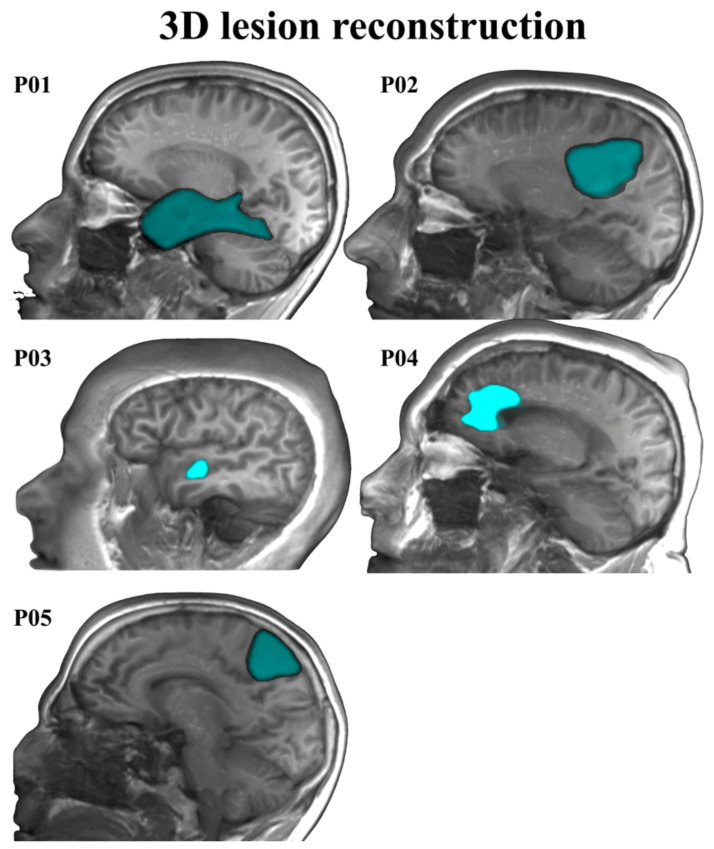
3D lesion reconstruction for the five patients.

**Figure 2 cancers-13-02593-f002:**
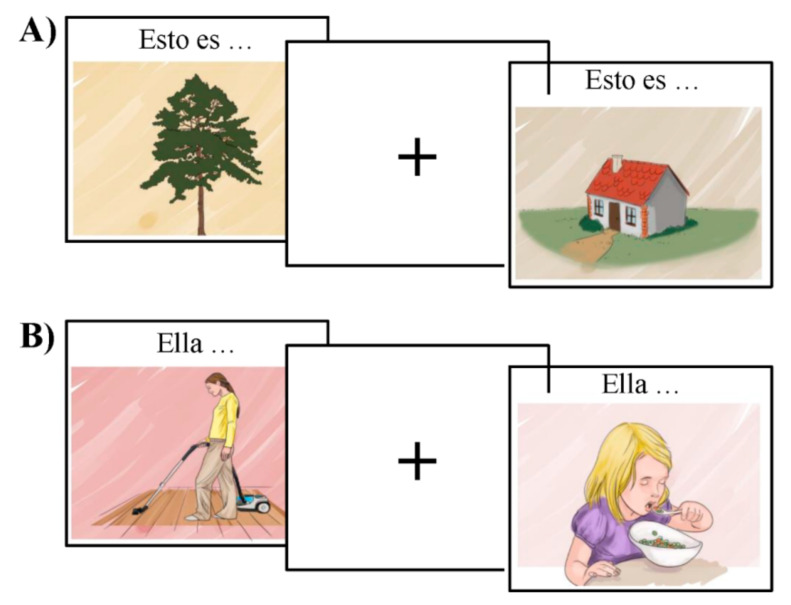
Example of the stimuli for the object in (**A**) and action naming task in (**B**).

**Figure 3 cancers-13-02593-f003:**
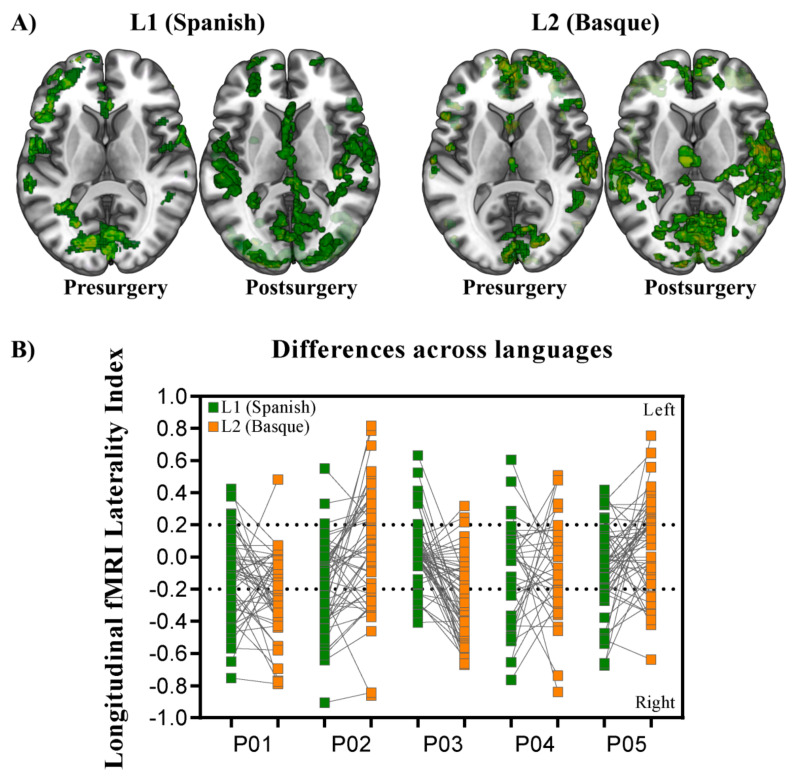
Group level representation. (**A**) Superimposition of the individual statistical parametric maps across participants. Each axial slice represents the superimposition resulting from each critical contrast (Language: Spanish and Basque; Time: pre- and post-surgery). MNI single-subject T1 image of MRIcroGL [58]) was used as template. (**B**) Chart representing longitudinal laterality changes for each of the five patients in L1 (Spanish) and L2 (Basque). Dotted lines represent the statistical threshold typically used for LI changes. Note that longitudinal laterality changes affect both L1 and L2.

**Figure 4 cancers-13-02593-f004:**
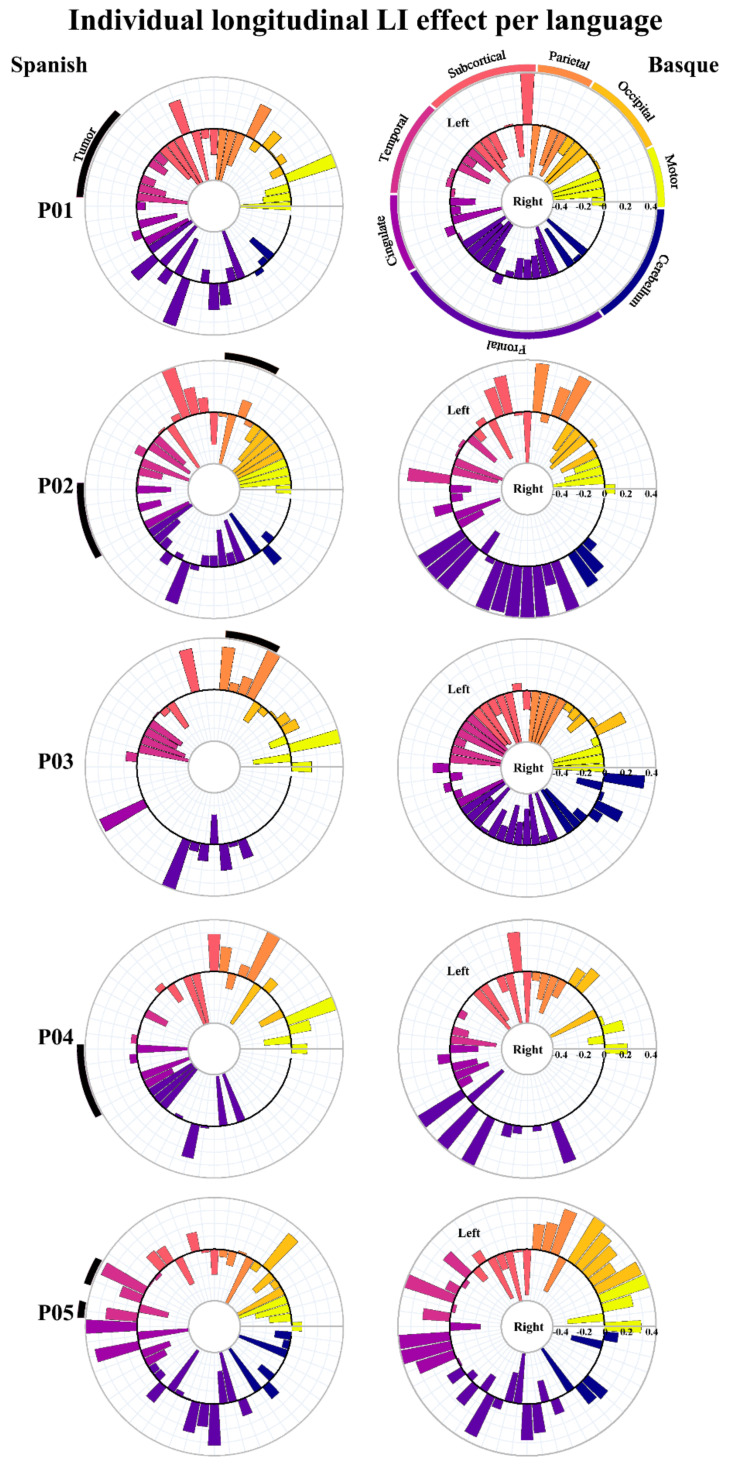
Charts representing laterality changes of the longitudinal effect for each of the five patients in Spanish and Basque. The middle circle represents zero laterality; bars going from the middle circle toward the outer circle represent increasingly left laterality; bars going from the middle circle toward the inner circle represent increasingly right laterality.

**Figure 5 cancers-13-02593-f005:**
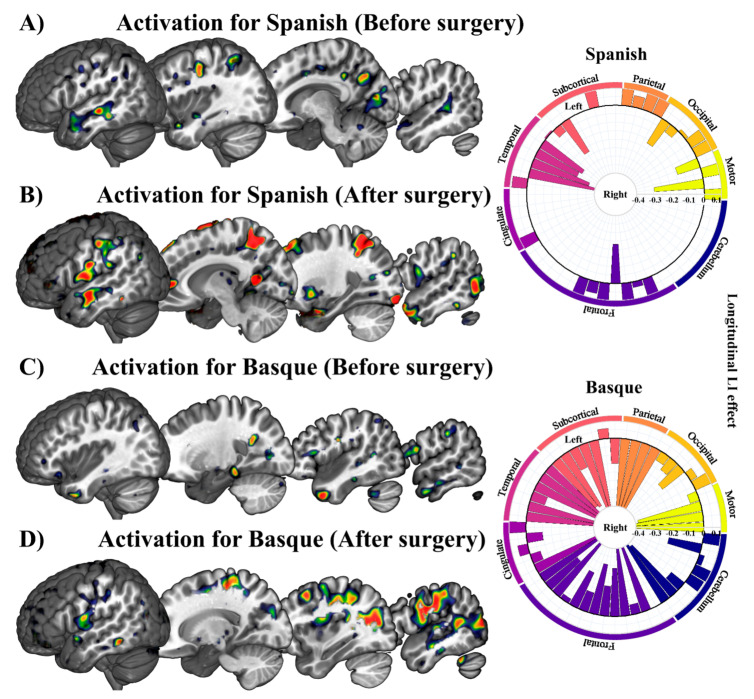
Statistical parametric maps resulting from each critical contrast in a prototypical patient with low-grade glioma. (**A**–**C**), and (**D**) represent Spanish and Basque before and after surgery, respectively. The chart represents laterality changes in the longitudinal effect for Spanish and Basque in different regions. The middle circle represents zero laterality; bars going from the middle circle toward the outer circle represent increasingly left laterality; bars going from the middle circle toward the inner circle represent increasingly right laterality.

**Figure 6 cancers-13-02593-f006:**
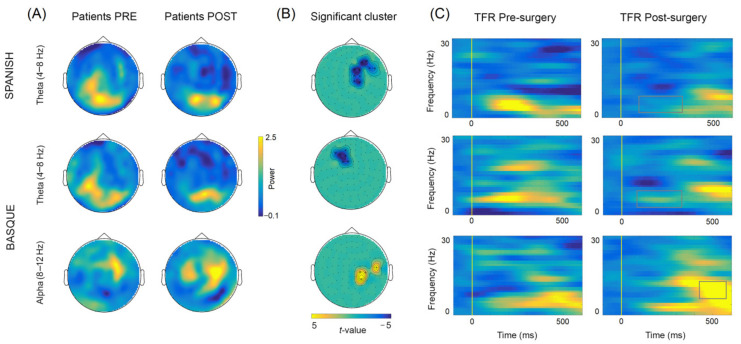
(**A**) Topographical power plots for the pre- and post-surgery stages (baselined with 500 ms pre-stimulus period) based on combined gradiometers averaged over the significant time-period within the theta (100 to 350 ms; 4 to 8 Hz) and alpha (350 to 500 ms; 8 to 12 Hz) bands. (**B**) Significant clusters showing longitudinal differences between conditions (post- vs. pre-surgery). (**C**) Mean time–frequency representations (TFRs) of the significant sensors within the clusters showing longitudinal power changes.

**Figure 7 cancers-13-02593-f007:**
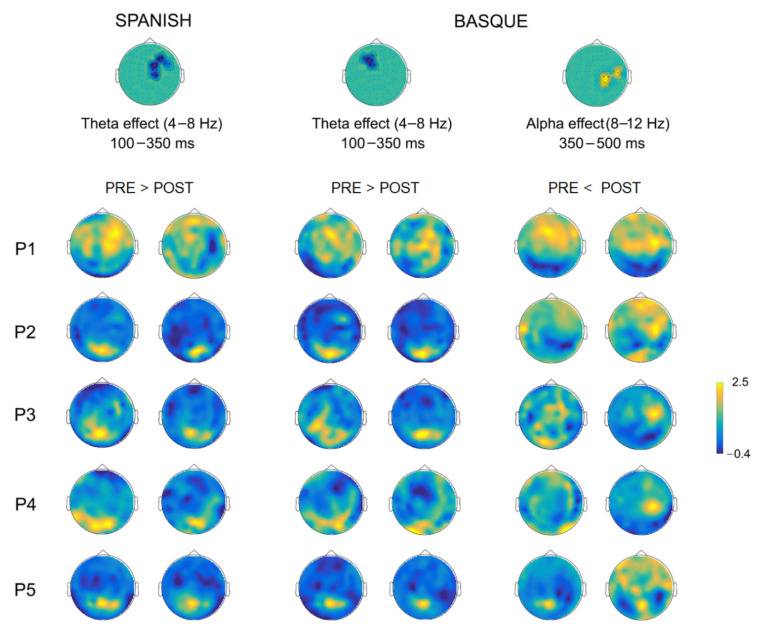
Individual topographical power plots for the pre- and post-surgery stages based on significant sensors (as shown by the clusters on top), time window, and frequency-band for each patient.

**Figure 8 cancers-13-02593-f008:**
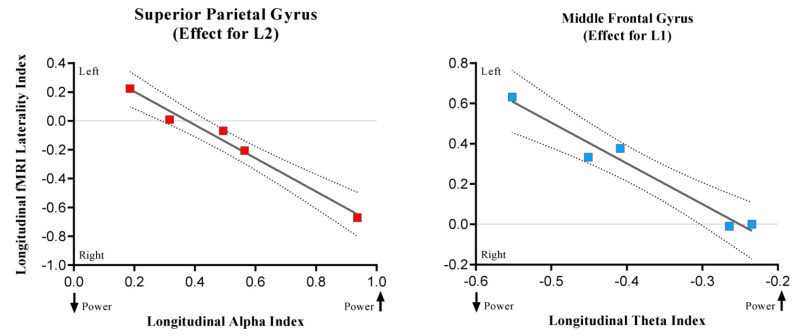
The scatterplots show significant associations between longitudinal fMRI lateralization indexes and oscillatory longitudinal power changes. While positive LI values indicate leftward lateralization, the negative ones indicate rightward lateralization. The arrows in the upper part of the charts indicate whether the power increases or decreases.

**Figure 9 cancers-13-02593-f009:**
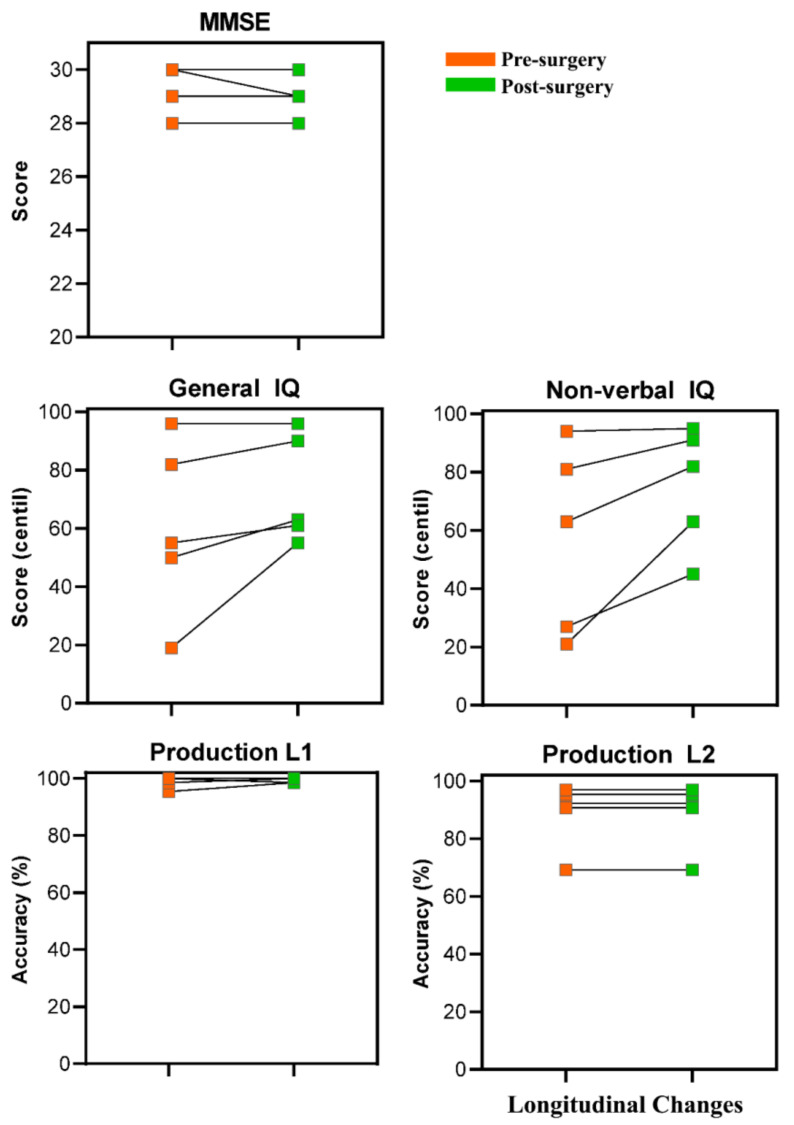
Charts representing the cognitive recovery of each patient in each of the three cognitive tasks (i.e., Minimental Cognitive State Examination, Kbit, and Picture Naming). Cognitive performance before and after tumor resection is shown for each patient in each task.

**Table 1 cancers-13-02593-t001:** Individual demographic features.

Patients	Age	Gender	Studies (Years)	Tumor Location	L1 Proficiency	L2 Proficiency	MMSE	Karnofsky Index
01	22	Male	14	Fusiform	95.38	92.31	30	90
02	47	Male	20	Cingulate	100	69.23	30	90
03	41	Male	20	Parietal	100	96.92	29	90
04	56	Male	12	Frontal	98.46	90.77	28	90
05	23	Male	16	STS	100	95.38	30	90

**Table 2 cancers-13-02593-t002:** Behavioral assessment.

Behavioral Tasks	Description
Spanish version of the Kaufman Brief Intelligence Test (KBIT) as a measure of verbal and non-verbal intelligence [34]	The verbal intelligence subtest is divided into two parts. The first part is a picture-naming test comprising 45 different items. The second part consists of 37 riddles, in which participants have to guess missing letters to reveal a hidden word. The non-verbal subtest measures the ability to solve new problems, detect relationships, and complete visual analogies.
Spanish version of the Mini-Mental State Examination (MMSE) as a measure of general cognitive status [33]	The MMSE is a 30-point screening test for evaluating cognitive impairments (30–27: normal; 26–25: possible cognitive impairments; 24–10: mild cognitive impairments; 9–0: moderate to severe cognitive impairments). It comprises different sections which assess spatiotemporal orientation, visuospatial attention, and language function.
Picture-Naming Test (BEST) as a measure of language production [35]	BEST is divided into two parts. The first part is a 65-picture naming test to be completed by participants in the two languages (i.e., Spanish and Basque). The second part is a short semi-structured interview guided by a multilingual linguist with experience in assessing language proficiency, who rates the participant’s skills in each language.

## Data Availability

The data presented in this study are available on request from the corresponding author. The data are not publicly available due to the data sharing policies of the different institutions involved.

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
