# Peer review of "What Can Glioma Patients Teach Us about Language (Re)Organization in the Bilingual Brain: Evidence from fMRI and MEG"

_cancers, 2021, doi:10.3390/cancers13112593_

Round 1

Reviewer 1 Report

The article by Quinones et al. describes a multimodal (fMRI/MEG) approach to examine language reorganization in bilingual individuals who have undergone tumor resection for a low grade glioma.  The authors not only found preserved language function utilizing these multimodal maps of eloquent cortex post-surgery, but functionally significant changes in neurophysiology (e.g. cortical reorganization) that inform how language representation shifts in bilingual patients.  While I found the approach valid and the findings intriguing, I feel that there are a few areas in the manuscript that need to be strengthened prior to acceptance.

While the goals of the investigation (and relevant studies) are clear in the introduction, the a priori hypotheses in the study do not seem to be perfectly clear.  Based on previous investigations, do the authors predict reorganization in L1/L2 representation?  Similarly, based on the current version of the introduction, the justification for using fMRI *and* MEG seems underdeveloped.  It’s common knowledge that the two complement each other with respect to spatial and temporal resolution, yet how do these strengths specifically contribute to our understanding of neuroplasticity?  The authors touch briefly on the importance of oscillations and cognition, but it does not seem transparent as to how this fits into the conceptual framework of the study. 

I feel that some of the articles in the Methods could be simplified.  For example, in Figure 1, there does not need to be a legend key for “T1 image” (this could be simply stated in the figure legend).  Table 2 outlines in great detail neuropsychological assessments that are commonplace in behavioral testing (like the MMSE) and could be either reduced or eliminated altogether. 

For fMRI, when were FDR vs. FWE thresholds applied?  Was the most conservative approach tried first, and then yielded to a more liberal threshold until results were obtained?

Non-phase locked induced changes in oscillatory activity (especially in low frequency bands like alpha/beta) are known to not only induce changes in lateralized language cortices in the temporal lobe but in lateralized frontal regions involved in speech production, in later time windows (e.g. Broca’s/IFG; see Findlay et al., 2012; Sowman et al., 2014; Ala-Salomaki et al., 2021).  Did the authors examine these later time windows?

Although the sensor-based MEG analyses are informative, did the authors try source based (inverse) solutions to reconstruct patterns of activation and putative shifts in cortical activity? 

Lines 295-307 : Although the investigators are certainly limited with the small sample (n=5) in the types of statistics they can use, and I approve the usage of a conjunction to show changes in the fMRI data, a lot of the statements made in this paragraph seem more observational than inferential.  When the authors say “significantly greater” (line 300) at what level (p-value, F statistic).  The same can be said when these authors describe changes in laterality (Figure 5), although presenting these as a case series and taking a descriptive approach is appropriate.

Although the small sample size of the study is certainly a limitation, given how well matched the participants are for L1/L2, tumor grade, etc. I feel that it is not a major issue but valuable that the study is presented as a case series, which the majority of the current manuscript follows.

Minor comments/questions

Line 45 : I don’t think it is accurate to state that “a capacity for neuroplasticity can be pathological”, implying that plasticity itself leads to pathology, which isn’t the case.  Plasticity can be induced in the presence of pathology (lesions), but does not generate the pathology itself.

Based on the description of the fMRI scan parameters, I’m assuming this is an event-related design?

Author Response

Reviewer 1

Comment 1. While the goals of the investigation (and relevant studies) are clear in the introduction, the a priori hypotheses in the study do not seem to be perfectly clear. Based on previous investigations, do the authors predict reorganization in L1/L2 representation? Similarly, based on the current version of the introduction, the justification for using fMRI *and* MEG seems underdeveloped. It’s common knowledge that the two complement each other with respect to spatial and temporal resolution, yet how do these strengths specifically contribute to our understanding of neuroplasticity? The authors touch briefly on the importance of oscillations and cognition, but it does not seem transparent as to how this fits into the conceptual framework of the study.

Response. We have taken time to meditate on this comment because all these points might strengthen the impact of the paper. The Introduction and Discussion of the manuscript have been modified in the new version highlighting the novelty of our multi-method approach and discussing how these findings could impact our understanding of neuroplasticity in the context of bilingual brain tumor patients. In addition, we have included a series of correlational analyses between the oscillatory response and the laterality fMRI indexes which highlighting the convergence but also the complementarity between both techniques (see paragraph 3, page 3, and figure 8).

Concerning the hypotheses of the study, we have included our specific predictions at the end of the Introduction as follows:

“Overall, based on previous literature, we hypothesized the existence of similar plasticity mechanisms as those observed in stroke patients (Bourdillon, Apra, Guenot, & Duffau, 2017; Hartwigsen, 2016, 2018; Hartwigsen et al., 2017; Hartwigsen & Saur, 2017), namely recruitment of ipsilesional healthy areas and contralateral homologs. Specifically, we expected that changes induced by these mechanisms would be evident in a longitudinal fashion (Amoruso et al., 2020). To the best of our knowledge, no study has previously tracked plastic changes in the bilingual brain through fMRI and MEG language mapping using a longitudinal approach. Nevertheless, based on studies investigating bilingual language control, we predicted a compensatory upregulation of domain-general areas involved in inhibition and attentional control mechanisms (Erika-Florence et al., 2014). As the superior parietal gyrus, the cingulate cortex, and the middle/superior frontal areas are at the heart of the inhibition and attentional control network, we specifically expected a language effect on some of these regions (Erika-Florence et al., 2014; Abutalebi & Green, 2016; van Dokkum et al., 2019), which might have a counterpart on cross-language alpha power changes (Obleser et al., 2012; Bice, 2020; Lizarazu et al., 2021).”

Comment 2. I feel that some of the articles in the Methods could be simplified. For example, in Figure 1, there does not need to be a legend key for “T1 image” (this could be simply stated in the figure legend). Table 2 outlines in great detail neuropsychological assessments that are commonplace in behavioral testing (like the MMSE) and could be either reduced or eliminated altogether.

Response. We agree with the reviewer regarding Figure 1. It was changed accordingly in the new version of the manuscript. Concerning Table 2, we simplify the description of the tasks. However, we think it is important to keep some details about the neuropsychological test to possible readers not familiarized with this kind of task (see Figure 1, and Table 2).

Comment 3. For fMRI, when were FDR vs. FWE thresholds applied? Was the most conservative approach tried first, and then yielded to a more liberal threshold until results were obtained?

Response. We acknowledge the Reviewer’s observation and agree about the lack of clarity in the description of this part. Individual effects were tested using a threshold of p < 0.001 uncorrected with a voxel extent higher than 50 voxels, then the statistical table was analyzed and the p and k values were corrected such that only those peaks or clusters with a p-value corrected for multiple comparisons using family-wise error rate (FWER; Nichols & Hayasaka, 2003) or false discovery rate (FDR; Genovese et al., 2002) were considered to be significant (p < 0.05). We have clarified this point in the new version of the manuscript (paragraph 1, page 8).

Comment 4. Non-phase locked induced changes in oscillatory activity (especially in low frequency bands like alpha/beta) are known to not only induce changes in lateralized language cortices in the temporal lobe but in lateralized frontal regions involved in speech production, in later time windows (e.g. Broca’s/IFG; see Findlay et al., 2012; Sowman et al., 2014; Ala-Salomaki et al., 2021).  Did the authors examine these later time windows?

Response. We acknowledge the Reviewer's observation and agree with him/her in the involvement of more frontal regions during speech production. Nevertheless, given the overt nature of our task, we focused on the initial 500ms period after picture onset to be sure that our TFR analysis was not contaminated by prearticular and/or articulatory muscle artifacts. Indeed, previous studies suggest that artifact-free recordings can be measured up to approximately ~400 ms post-stimulus presentation (Ganushchak, Christoffels, & Schiller, 2011). Please note that for instance in the case of Ala-Salomaki et al., 2021 study they specifically requested participants to wait and give their overt response after a short delay period (i.e., 1000ms) to avoid motor artifacts. In our case, participants gave their responses right after the presentation of the picture, thus based on reaction times of participants and visual inspection of the signal, we decided to circumscribe our analysis to a “safe” time window.

Comment 5. Although the sensor-based MEG analyses are informative, did the authors try source based (inverse) solutions to reconstruct patterns of activation and putative shifts in cortical activity?

Response. We acknowledge the Reviewer's remark. However, given the fact that we had the spatial information provided by the fMRI data, which is arguably more accurate than the MEG one; we decided to focus on the advantages provided by each technique. Furthermore, previous studies show some inconsistencies between fMRI and MEG responses during picture naming, potentially due to time-related aspects (https://pubmed.ncbi.nlm.nih.gov/19378277/). However, if the Reviewer deems it necessary we are more than pleased to perform the analysis and include it in a new version of the manuscript.

Comment 6. Lines 295-307 : Although the investigators are certainly limited with the small sample (n=5) in the types of statistics they can use, and I approve the usage of a conjunction to show changes in the fMRI data, a lot of the statements made in this paragraph seem more observational than inferential.  When the authors say “significantly greater” (line 300) at what level (p-value, F statistic). The same can be said when these authors describe changes in laterality (Figure 5), although presenting these as a case series and taking a descriptive approach is appropriate.

Although the small sample size of the study is certainly a limitation, given how well matched the participants are for L1/L2, tumor grade, etc. I feel that it is not a major issue but valuable that the study is presented as a case series, which the majority of the current manuscript follows.

Response. We acknowledge the Reviewer for this important comment and we agree with his/her opinion about how the methodological issues concerning the sample size negatively affect the study of pathological populations. However, to minimize these effects, we analyzed fMRI data following a single case approach and that is why we presented the statistical results of one prototypical participant in Figure 5. This figure includes the statistical parametric maps resulting from the four critical contrasts. Only those significant clusters corrected for multiple comparisons were depicted in the figure. In order to simplify the Result session, we represented only one participant. In the same way, the laterality index analyses were performed using the same single case approach. We employed a threshold independent bootstrapping method per participant per region following the procedure included in the Laterality Index SPM Toolbox (Wilke & Lidzba, 2007). LI was estimated as LI = (left − right)/(left + right), resulting in positive values for left-dominance and negative values for right-dominance (Bradshaw, Bishop, & Woodhead, 2017). This involves iterative resampling and estimation of LIs across multiple threshold levels for all possible right/left sample combinations. To reduce the effect of outliers, trimmed means taken from the middle 50% of the t-value distribution were used as the final LI scores (Wilke & Lidzba, 2007). Lateralization indexes spanned the full range of possible values between -1 (fully right-lateralized) and 1 (fully left-lateralized). Following the procedure described in Wilke & Lidzba (2007) LI values higher than 0.2 and lower than -0.2 are considered significant.

Comment 7. Line 45 : I don’t think it is accurate to state that “a capacity for neuroplasticity can be pathological”, implying that plasticity itself leads to pathology, which isn’t the case.  Plasticity can be induced in the presence of pathology (lesions), but does not generate the pathology itself.

Response. We acknowledge the Reviewer for this comment and we strongly agree with this point. The sentences were rephrased as follows: “A remarkable feature of the central nervous system is its inherent capacity to dynamically reorganize its structure and function depending on the environment (Carreiras et al., 2009; Duffau, 2014a; Gurunandan, Arnaez-Telleria, Carreiras, & Paz-Alonso, 2020; Gurunandan, Carreiras, & Paz-Alonso, 2019; Herbet, Maheu, Costi, Lafargue, & Duffau, 2016; Oliver, Carreiras, & Paz-Alonso, 2016). This capacity for neuroplasticity can give rise to changes in both healthful (e.g., acquisition of a new language; Berken, Gracco, & Klein, 2017; Friederici, 2017)) and pathological (e.g., presence of brain lesions; Duffau, 2014a; and Herbet et al., 2016) contexts.”

Comment 8. Based on the description of the fMRI scan parameters, I’m assuming this is an event-related design?

Response. Yes, it is. Each experimental session was planned following an event-related design where we alternated verbs or nouns with different lexical properties. For the purpose of the current study, we analyzed the different conditions together without making any assumptions about the possible differences across conditions. This point has been clarified in the current version of the manuscript.

Reviewer 2 Report

The authors examined language reorganization in bilingual individuals with low grade glioma. They examined whether the tumor and its resection affected reorganization differently for the two languages in these individuals. They examined language maps of object naming and verb generation tasks derived using fMRI and MEG in five bilingual patients before and after tumor resection.

Overall the manuscript is well written and the data are presented cogently. I have a few minor comments.

  1. Were all 4 tasks (Object naming in Spanish and Basque and Verb generation in Spanish and Basque) acquired in one scan? Although the tasks were performed as event related, were the naming and verb generation tasks performed as blocks or were they randomly presented? If it is the latter case, could switching between tasks add additional cognitive burden during task performance and result in activation of the regions implicated in attention and executive functions.
  2. Could the authors expand on what is the possible relationship between MEG findings and fMRI findings? For instance, how do the increased activity noted inf MRI relate to desynchronization and/or hypersynchronization found in MEG?
  3. Finally, the authors conclude that in all patients, surgical intervention and intra-operative language mapping contributed to successfully preserving cortical and subcortical structures critical for processing language functions. With this regard, it is not clear how the presurgical fMRI and MEG mapping and the intraoperative mapping were used to inform surgical procedure. Also there is no data on resection of language areas (if any) in the individual patients. It will be helpful to ascertain this if the authors can provide these information.

Author Response

Reviewer 2

Comment 1. Were all 4 tasks (Object naming in Spanish and Basque and Verb generation in Spanish and Basque) acquired in one scan? Although the tasks were performed as event related, were the naming and verb generation tasks performed as blocks or were they randomly presented? If it is the latter case, could switching between tasks add additional cognitive burden during task performance and result in activation of the regions implicated in attention and executive functions.

Response. As the reviewer pointed out, switching between object naming and verb generation adds additional cognitive burden during task performance and this might result in the activation of areas implicated in attention and executive functions. Thus, in order to avoid this confound, object naming and verb generation were tested in separate sessions. Each experimental session (1. Object naming in Spanish; 2. Object Naming in Basque; 3. Verb generation in Spanish and 4. Verb generation in Basque) was planned following an event-related design where we alternated verbs or nouns with different lexical properties. For the purpose of the current study, we analyzed the different conditions together without making any assumptions about the possible differences across conditions. This point has been clarified in the current version of the manuscript.

Comment 2. Could the authors expand on what is the possible relationship between MEG findings and fMRI findings? For instance, how do the increased activity noted inf MRI relate to desynchronization and/or hypersynchronization found in MEG?

Response. We thank the Reviewer for bringing this important aspect to our attention. We have now performed a series of correlational analyses to link results from both techniques. First, we calculated the mean of theta and alpha power within the significant sensors, and time windows highlighted by the cluster-based permutation test in each language (i.e., Spanish and Basque) and stage (i.e., before and after tumor resection). Second, we estimated power indexes by subtracting post- and pre-surgery mean values, to capture the longitudinal effects highlighted by the clusters. Finally, these indexes were correlated with the longitudinal laterality indexes derived from the fMRI analysis. Pearson correlations between MEG and fMRI longitudinal measures were performed. Bonferroni correction was applied to control for the probability of committing a type I error adjusting the alpha value for the number of statistical tests. Interestingly, we found specific associations between BOLD laterality indexes and alpha and theta activity (see Figure 8). This information has been added to the manuscript and the discussion session has been modified accordingly.

Comment 3. Finally, the authors conclude that in all patients, surgical intervention and intra-operative language mapping contributed to successfully preserving cortical and subcortical structures critical for processing language functions. With this regard, it is not clear how the presurgical fMRI and MEG mapping and the intraoperative mapping were used to inform surgical procedure. Also there is no data on resection of language areas (if any) in the individual patients. It will be helpful to ascertain this if the authors can provide these information.

Response. We acknowledge the Reviewer for this important question. The main goal of presurgical imaging (fMRI-MEG) is to guide the design of the intraoperative testing strategy. The specific selection of tasks that will be performed during surgery is oriented and pre-tested in fMRI and MEG to choose the ones more "apriori" useful for the case taking of course into account the anatomical location of the lesion. These imaging techniques not also inform about the possible precise location of functions around the lesion, but also its dynamics plasticity and they are a window to see the whole-brain functioning. This is complementary to the intraoperative functional results of direct brain stimulation, which only gives practical information for the resection as they only provide information about the exact point of stimulation, not about the whole network. Then, our functional limits will be set by intraoperative stimulation itself and its results, when applied to these specific testing designed and tailored based on fMRI and MEG. We do not perform the surgical resection based on imaging but based on the patient's clinical response. This is not a functionally imaging-guided surgery. Functional limits are all established with the clinical intraoperative response, cortical and subcortical, and they are always respected if a 3-time stimulation spot is positive. We do not resect any cortex or fascicle with a positive response. For a better understanding of how these functional limits are taken into account see the following examples:

-Gil-Robles, S., & Duffau, H. (2010). Surgical management of World Health Organization Grade II gliomas in eloquent areas: the necessity of preserving a margin around functional structures. Neurosurgical focus, 28(2), E8.

-Robles, S. G., Gatignol, P., Lehéricy, S., & Duffau, H. (2008). Long-term brain plasticity allowing a multistage surgical approach to World Health Organization Grade II gliomas in eloquent areas: report of 2 cases. Journal of Neurosurgery, 109(4), 615-624.

Round 2

Reviewer 1 Report

I am very satisfied with the changes made by the authors in the manuscript, particularly with the structure of the manuscript, hypotheses posed and presentation of results.  The addition of correlations between fMRI and MEG findings are also intriguing, but do raise new questions.    I have a few minor points that need to be addressed before acceptance.
1)  It's still not entirely transparent (at least to me) how multiple comparisons corrections were applied.  The authors state in the revisions that clusters that met either FWER *or* FDR thresholding levels were reported as statistically significant, but this is confusing, as is selection criteria for when FWER or FDR were applied.  FWER is known to be notoriously conservative (with respect to FDR).  There should not be a scenario where a cluster appears significant at FWER if it fails to be significant at  a FDR threshold.  There should, however, be substantial overlap in the table between what is significant with FDR when it is significant at FWER.  Was this the case?  
2)  The correlations between LI and alpha/theta power change are certainly compelling (and near perfect linear relationships) although might be a little too ambitious in their claims (that right hemisphere reorganization is driven by increased longitudinal alpha power in SPL and decreased longitudinal theta power in MFG).  Generating such focal inferences from sensor based MEG data (well known for signal leakage, bleed and lack of spatial specificity) can be problematic.  How confident are the authors that the changes are from the SPL/MFG, and not the result of spatial blur from the sensors?  It would be useful to have more clarification here methods-wise, or instead correlate activation with specific sensors from the montage they are including (and then speculate as to what cortical fields are generating that signal).  Circling back to my previous reviews, this would be something inverse solutions of the data would be better at addressing/localizing.
